# Interrogating a framework for diabetic retinopathy screening adherence: Qualitative insights from a severely affected and under-adherent population

Julia Fu[1], Joana Andoh[2], Elizabeth Fairless[3], June Weiss[4], Althea Norcott[5], Kristen Nwanyanwu[1]*

**1** Department of Ophthalmology and Visual Science, Yale School of Medicine, New Haven, Connecticut, United States of America, **2** Department of Ophthalmology, Wilmer Eye Institute, Johns Hopkins School of Medicine, Baltimore, Maryland, United States of America, **3** Department of Ophthalmology and Visual Science, Moran Eye Center, University of Utah, Salt Lake City, Utah, United States of America, **4** Department of Internal Medicine, Yale School of Medicine, New Haven, Connecticut, United States of America, **5** Community Contributor, New Haven, Connecticut, United States of America

* k.nwanyanwu@yale.edu

## Abstract

To validate and expand upon a framework of diabetic retinopathy (DR) screening adherence by examining barriers and facilitators among individuals with severe DR and those under-adherent to screening. From March 2021 to February 2022, we conducted eight remote semi-structured interviews with adults with diabetes across two participant populations: (1) participants with severe DR who had undergone procedures (n = 4) and (2) participants under-adherent to screening, defined as no eye exam in >1 year (n = 4). We recruited participants from the Yale Eye Center and community referrals. During the interviews, we collected demographic data and presented participants with a DR screening adherence framework previously developed by our group. We transcribed all interviews and conducted analyses using a hybrid deductive-inductive analytic approach informed by grounded theory techniques to identify recurring themes. Themes across both participant populations aligned with the existing framework, including *vision status*, *emotional context*, *competing concerns*, *resource availability*, *cues to action*, *knowledge-creating experiences*, *and in-clinic experiences*. The *patient-doctor relationship* emerged as a sub-theme of *in-clinic experiences*, highlighting the role of trust and communication in supporting sustained eye care engagement. At the individual level, participants with stable vision often perceived no need for screening. At the institutional and structural level, participants identified lack of insurance and transportation as significant barriers. These findings support the robustness of the DR screening adherence framework across varying levels of disease severity and engagement. Interventions that improve patient education, address structural barriers, and strengthen patient-doctor relationships may enhance screening adherence among populations at high risk for vision loss.

**Data availability statement:** All relevant data are within the manuscript and its Supporting information files.

**Funding:** This publication was made possible by Grant Number 1 K23 EY030530-01 from the National Eye Institute (K.N.) and an unrestricted/challenge award to Yale Eye Center from the Research to Prevent Blindness (RPB), Inc (K.N.). The funders had no role in study design, data collection and analysis, decision to publish, or preparation of the manuscript. No authors received a salary from any funders.

**Competing interests:** The authors have declared that no competing interests exist.

## Introduction

Diabetic retinopathy (DR) is a leading cause of blindness among working-age adults in the United States with approximately 9.6 million people currently living with DR [1]. Current guidelines recommend at least annual eye screenings for individuals with diabetes, which may aid in early detection and prevention of vision loss [2,3]. However, it is estimated that only 56.9% of the more than 93 million US adults at high risk for vision loss receive eye care annually [4]. Therefore, it is crucial to understand the factors influencing DR screening adherence and develop relevant interventions to improve access.

Previous studies have explored the barriers to DR screening adherence through qualitative methods, which enable a deeper understanding of individuals' diverse experiences [5]. Hartnett et. al. conducted focus groups with individuals with diabetes and identified key barriers to eye care such as financial difficulties and limited access to services [6]. Similarly, Elam et. al. conducted focus groups with individuals at high risk for vision loss and found that common barriers to eye care included high costs, lack of trust in healthcare providers, clinic inaccessibility, and poor patient-doctor relationships [7].

In the United States, Hispanic and African American populations are disproportionately affected by DR and may be less likely to receive DR screening [8,9]. Additionally, studies have reported lower awareness of DR diagnosis among these populations, which can further limit engagement in eye care [10]. Social and cultural factors may also influence engagement in eye care. For example, cultural norms emphasizing self-reliance or managing health concerns independently may lead some individuals to delay or forgo professional eye care, particularly if symptoms are mild [11]. Additionally, limited cultural competency in providers, including lack of awareness of health beliefs or appropriate communication styles, may reduce trust and deter individuals from seeking care [11].

Our research team previously developed a framework to understand DR screening adherence by conducting qualitative interviews with individuals with diabetes at high-risk for vision loss [12]. This framework outlined the individual and institutional factors affecting DR screening adherence. Individual factors included vision status, competing concerns, and the emotional context of receiving screenings. Institutional factors included resource availability, cues to action, knowledge-creating experiences, and in-clinic experiences.

In this study, we aim to validate and expand upon the previously developed DR screening adherence framework by examining barriers and facilitators among two participant populations: participants with severe DR and participants under-adherent to DR screening. By applying the framework across participants with differing levels of disease severity and engagement, we aimed to assess the robustness of the framework and identify additional factors influencing screening behavior.

## Methods

### Ethics statement

The institutional review board (IRB) of Yale University approved this study. This study abided by the tenets of the Declaration of Helsinki. Leaders of local community health

organizations through the Yale Community Engaged Research Steering Committee advised the study design. Prior to each interview, we obtained verbal informed consent for participation in the study and documented consent via recording and transcription of the interview. We did not obtain written informed consent due to the remote nature of the interviews. The IRB of Yale University approved our use of verbal consent.

Between March 9th, 2021, and February 17th, 2022, we recruited participants through the Yale Eye Center and referrals from community partners. As part of recruitment, we contacted participants by phone or email to introduce the study and gauge their interest. The researchers had no prior relationships with the participants. We selected participants using maximum variation sampling, with participants purposively selected to capture variation in engagement patterns and disease severity.

We conducted eight individual semi-structured, qualitative interviews in English with adults diagnosed with diabetes. We recruited four of these participants through the Yale Eye Center and four through referrals from community partners. To capture more diverse participant experiences, we interviewed participants from two participant populations: (1) participants with severe DR (n = 4) and (2) participants under-adherent to DR screening, defined as no eye exam in the past year (n = 4). Participants with severe DR included individuals with clinically documented DR who had previously undergone laser treatment or other ocular procedures. We did not select these participants based on engagement behavior, though a history of ocular interventions indicates some degree of prior interaction with eye care. However, this does not necessarily imply ongoing or regular engagement. Under-adherent participants included individuals with diabetes who had not sought or obtained an eye exam for the past year, irrespective of the underlying reason. This does not imply that these participants never sought eye care but rather reflects a lack of routine engagement while still enabling symptom-driven care. We classified this cohort solely by recent engagement behavior and independently of DR status.

Study author J.A. (medical student), who identifies as an African American woman, conducted the interviews by phone or web-based videoconferencing software (Zoom, San Jose, CA) and recorded the interviews for transcription. J.A. trained in qualitative methods through institutional modules and the textbook, *Qualitative Research: A Guide to Design and Implementation (*4th Edition, ISBN: 978-1-119-00361-8). Only participants and researchers were present during the interviews.

At the start of each interview, we introduced participants to the researcher, including her educational background and interest in the topic, and shared the purpose of the study. We then collected basic demographic information and presented participants with a slide deck (S1 Text) containing the thematic framework previously developed by our research group related to DR screening barriers [12]. Interviews were guided by a flexible interview guide with open-ended prompts (Table 1). Each interview lasted approximately 60 minutes.

We transcribed all interviews for analysis. We analyzed transcripts using a hybrid deductive-inductive analytic approach informed by grounded theory techniques, particularly constant comparison (S2 Text, S3 Text, S1 Table) [13]. Coding began with a priori codes from our existing framework and allowed for new themes to emerge [12]. Three authors (J.A., K.N., and A.N., a trained community partner) independently reviewed transcripts line by line. The study team (K.N., J.A., and A.N.) held two collaborative sessions to iteratively review transcripts, refine codes and interpretations, and adjust subsequent interviews based on emerging insights. We achieved code saturation within these narrowly defined cohorts when no new codes emerged across successive interviews. We contextualized findings using the socio-ecological model (SEM), examining individual, interpersonal, and structural influences on care-seeking behavior. Following the interview, participants were mailed a $20 gift card and received a slide deck summarizing study findings. They were invited to provide feedback during the session. No repeat interviews were conducted, and transcripts were not returned to participants for correction. Analysis was supported by NVivo software, version 12 (Melbourne, Australia).

## Results

We conducted interviews with four participants with severe DR and four participants classified as under-adherent. Participants with severe DR were 75% male with an average age of 58.5 years (range 52–64). Under-adherent participants were

**Table 1. Interview Guide.**

| |
|---|
| **Resource Availability** |
| Has anything ever prevented you from having an eye exam? |
| Have you ever cancelled or not shown up to an appointment? If so, why? |
| Is there anything that would make it easier for you to get an eye exam? |
| Who or what has helped you stay on top of your eye exams? |
| If barrier X was removed, what do you think would happen? |
| **Cues to Action** |
| What made you decide to get/not get a dilated eye exam? |
| What brought you here today? What brought you to your last eye exam? |
| What factors motivated you? What factors reminded you? |
| Has anyone encouraged you to get an eye exam? |
| Were any factors more important than others? |
| **Knowledge-creating Experiences** |
| How did you learn about eye exams? |
| How did you know you needed one? |
| What factors helped you to know? |
| At the time that you were diagnosed with diabetes, what, if anything, were you told about eye care? |
| How did you learn that diabetes can affect your eyes? |
| Have you ever been told by a healthcare provider that diabetes can affect your eyes? |
| **In-Clinic Experience** |
| Can you tell me about your experience the last time you had an eye exam where your pupils were dilated? |
| Can you walk me through your experience? Start with the events leading up to the exam. |
| What was the experience like? What did you think then? |
| Did you encounter problems at any point? |
| What would have made your experience better? |
| How would you describe your relationship with your eye doctor? |
| **Vision Status** |
| When was the first time you noticed changes in your vision? |
| What about your vision has changed recently? |
| What changes startled or concerned you the most? |
| **Competing Concerns** |
| Have you ever cancelled or not shown up to an appointment? If so, why? |
| If you can remember, what happened? |
| What would make you reschedule or cancel an eye exam? |
| **Emotional Context** |
| When was the last time you had a dilated eye exam? |
| Tell me about how often you get an eye exam. |
| How often do you think you should have an eye exam? |
| If there's a difference between those two answers, why? |
| How did making an appointment make you feel? |
| How did getting an eye exam make you feel? |
| What would you have changed about the experience of getting an eye exam? |

50% male with an average age of 61.8 (54–71). Participants with severe DR were diagnosed with diabetes on average 24.9 years ago (range 3–36), while under-adherent participants were diagnosed on average 7.5 years ago (range 4–10).

The average time since the last eye exam was 109.3 days (range 6–365) for participants with severe DR, and 840.5 days (range 366–1825) for under-adherent participants. Additional participant demographic details are provided in Table 2.

Across both participant populations, recurring themes aligned closely with our previously developed framework, including *vision status, emotional context, competing concerns, resource availability, cues to action, knowledge-creating experiences*, and *in-clinic experiences* (Fig 1). The consistency of these themes across participants with differing levels of disease severity and engagement supports the robustness of the framework. In addition, the *patient-doctor relationship* emerged as a distinct sub-theme within *in-clinic experiences*, further refining the framework by highlighting the importance of longitudinal trust and interpersonal relationships in supporting sustained engagement in DR screening.

We classified these themes into two levels based on the SEM: 1) Individual level and 2) Institutional and Structural level. As in our previous study [12], the individual level includes *vision status, competing concerns, and emotional context*. The institutional and structural level includes *resource availability, cues to action, knowledge-creating experiences, and in-clinic experiences* with a sub-theme of the *patient-doctor relationship*, which was newly identified in this study.

## 1) Individual level

**Vision status.** Participants reported that changes in vision status including the need for new glasses influenced their adherence to DR screening. One participant with severe DR explained: "I'm just trying to keep up with new frames," (Participant 2, Severe DR). Similarly, an under-adherent participant said: "I need to change my eyeglasses, because the ones I have, I've had those [for] about eight years," (Participant 4, Under-adherent).

**Table 2. Participant demographic information.**

|  | Original Cohort | Participants with Severe DR | Under-adherent Participants |
|---|---|---|---|
|  | n = 24 | n = 4 | n = 4 |
| **Age, average (range)** | 57.7 (44-73) | 58.5 (52-64) | 61.8 (54-71) |
| **Gender, No. (%)** |  |  |  |
| Male | 14 (58) | 3 (75) | 2 (50) |
| Female | 10 (42) | 1 (25) | 2 (50) |
| **Race, No. (%)** |  |  |  |
| Black/African American | 14 (58) | 3 (75) | 4 (100) |
| White | 4 (17) | 0 | 0 |
| More than 1 race | 0 | 1 (25) | 0 |
| Other/No answer | 6 (25) | 0 | 0 |
| **Ethnicity, No. (%)** |  |  |  |
| Hispanic/Latino/a/x | 3 (13) | 1 (25) | 0 |
| Non-Hispanic/Latino/a/x | 21 (88) | 3 (75) | 4 (100.0) |
| **Insurance Status, No. (%)** |  |  |  |
| Insured | 23 (96) | 4 (100) | 4 (100) |
| Uninsured | 1 (4) | 0 | 0 |
| **Eye Exam Frequency, No. (%)** |  |  |  |
| Annually or more frequently | 17 (71) | 4 (100) | 0 |
| Biennially or less frequently | 7 (29) | 0 | 4 (100) |
| **Duration of diabetes, No. (%)** |  |  |  |
| 5 years or less | 8 (33) | 1 (25) | 1 (25) |
| 10 years or less | 6 (25) | 0 | 3 (75) |
| More than 10 years | 10 (42) | 3 (75) | 0 |

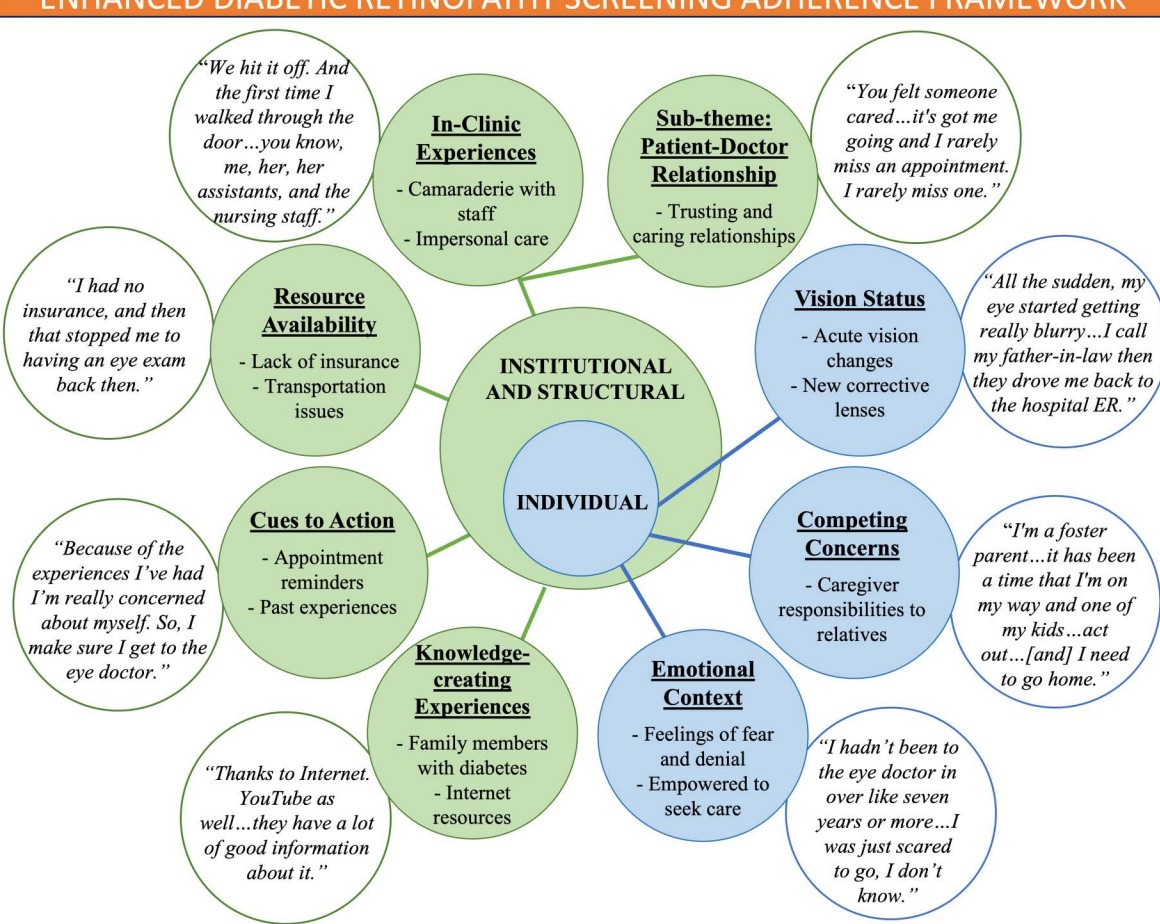

**Fig 1. Enhanced diabetic retinopathy screening adherence framework integrating themes identified across two participant populations: individuals with severe diabetic retinopathy and individuals without an eye exam in the past year.** The *Patient-Doctor Relationship* emerged as a refined sub-theme of *In-Clinic Experiences*.

Acute vision loss also prompted participants to seek eye care. A participant with severe DR described how a sudden change in vision led to an emergency eye exam: "All of a sudden, my eye started getting really blurry…I called my father-in-law, [and] they went and picked me up, then they drove me back to the hospital emergency room," (Participant 3, Severe DR). Other participants sought care due to symptoms like twitching, seeing blood vessels in their eyes, or experiencing transient blurry vision. One under-adherent participant was prompted to seek care after noticing a shadow in her vision: "I could see a shadow coming around my eye. When I got it, I called for Yale," (Participant 4, Under-Adherent).

Participants did not perceive a need for eye screening if their vision remained stable. A participant with severe DR stated: "I did not pay so much attention to that because I have 20/20 vision…I was not rushed or worried to go back to the treatment, because like I said, I did not understand how bad my eye was," (Participant 3, Severe DR). Similarly, an under-adherent participant remarked: "If I got to get eye appointments, then usually it'll be something that motivates me to…blurry vision or something like that," (Participant 1, Under-Adherent).

Though some participants described seeking eye care in response to acute symptoms or functional needs such as obtaining new glasses, these episodes did not necessarily translate into routine screening. Rather, this engagement was more reactive and therefore intermittent, occurring only when vision changes became subjectively noticeable.

**Competing concerns.**  Participants reported that competing concerns prevented them from accessing DR screening. A participant with severe DR explained: "I'm a foster parent. And, you know, it has been a time that I'm on my way and one of my kids acts out, you know, and so I'm in the car [and] I need to go home," (Participant 1, Severe DR). Similarly, an under-adherent participant said: "I just was going, then suddenly, I stopped when my husband was sick and couldn't get around. I went when I could, but if I had to take him or let him go to the doctor or something, I didn't have a way," (Participant 4, Under-Adherent).

**Emotional context.**  Fear and denial influenced screening adherence among participants. One participant with severe DR reflected on their long delay seeking care due to fear: "I hadn't been to the eye doctor in over like seven years or more. I was just scared to go, I don't know. I don't know why. I guess I didn't want to know what was going on with my eye," (Participant 4, Severe DR). Similarly, an under-adherent participant said: "It kind of scares me a little bit because I don't want to go blind. I want to see. I don't want nobody leading me around, you know…that's the wake-up for me too," (Participant 4, Under-Adherent).

Other participants felt empowered to seek care. One participant with severe DR stated: "Well, it really didn't bother me. It takes a lot to really get me riled up about anything. I took care of it. It was rectified. And I know what to do," (Participant 2, Severe DR). Another participant with severe DR expressed acceptance of their condition: "I'm not afraid to be blind at this point. Honest to you. Like before, I was afraid to be blind…I don't want that to happen, but if it happens, it happens," (Participant 3, Severe DR).

Participants expressed fear related to vision loss as both a motivator and a barrier to DR screening adherence. For some, fear initially delayed care but later transformed into sustained engagement through acceptance of their condition and empowerment to seek care. For others, fear did not consistently translate into routine screening. These findings suggest that emotional responses alone may not be sufficient to ensure adherence. Rather, the ability to access screening may depend on the interaction between the emotional context of eye care and structural or interpersonal supports.

## 2) Institutional and structural level

**Resource availability.**  Participants reported lack of insurance as a major barrier to accessing eye care. One under-adherent participant said: "One time I didn't have insurance. That stopped me a lot, and I had to pay out of my own pocket," (Participant 4, Under-Adherent). Another participant with severe DR said: "I had no insurance, and then that stopped me from having an eye exam back then. And then from there to'18, that's when the problems occurred in my eye," (Participant 3, Severe DR). Lack of transportation also emerged as a significant barrier. One under-adherent participant said: "Well, just I wasn't able to get there, that's all. If I would have had somebody to come bring me, I would have been there. And then I have to go through the stress of asking somebody to bring you, have to deal with their attitude and whatever they were supposed to be doing at the time. I don't want to go through that. That's why I didn't make it," (Participant 3, Under-Adherent). Another participant with severe DR expressed: "I even used to take the bus, okay, I was at work and had an eye doctor appointment. I had no car, couldn't drive because I couldn't see," (Participant 4, Severe DR).

Housing insecurity also impacted access to care for some participants. One under-adherent participant shared: "I'm homeless as we speak but I have somewhere to be at for now. But I am definitely homeless," (Participant 3, Under-Adherent). When asked how housing insecurity affected eye care, they said: "It's difficult, but I manage," (Participant 3, Under-Adherent). Although this participant did not elaborate on the specific challenges, housing insecurity may precipitate barriers to eye care such as lack of a primary care provider, low health literacy, financial constraints, and lack of transportation [14,15]

**In-clinic experiences.**  Participants described how in-clinic experiences influenced their adherence to screening. Some participants highlighted positive interactions with staff. One participant with severe DR shared: "We hit it off. And the first time I walked through the door, we had, you know, me, her, her assistants, and the nursing staff. Very good. Nothing bad

to say about any of them," (Participant 2, Severe DR). One under-adherent participant noted: "When I go there, everybody remembers me," (Participant 4, Under-Adherent). Another under-adherent participant appreciated the clear guidance provided during visits: "They're very informative. They tell you about whatever you needed prior to, and they walk you through it. So, there isn't any mystique to it," (Participant 2, Under-Adherent). However, participants also reported negative experiences. One participant with severe DR expressed feeling dehumanized: "I felt like I was a number. I was another number, not even a patient," (Participant 3, Severe DR). Another participant with severe DR echoed this sentiment, emphasizing a desire for more personal treatment: "You wanted to be treated as a person. So, that's the only problem I had," (Participant 4, Severe DR).

**Patient-doctor relationship**: A key sub-theme within the broader theme of *In-Clinic Experiences* was the *Patient-Doctor Relationship*. Positive interactions with physicians played a significant role in promoting screening adherence. One participant with severe DR remarked: "You felt someone cared…that's what brought me home and it's got me going and I rarely miss an appointment. I rarely miss one," (Participant 1, Severe DR). Another participant with severe DR noted appreciation of their doctor's caring attitude and clear communication: "She has the best attitude…she takes the time to talk to me and make me understand the problem I'm going through," (Participant 3, Severe DR). Another participant with severe DR emphasized the positive relationship they had with their doctor: "Why would I not go for an eye doctor's appointment where I love the doctors?...I told whoever my insurance people were, I want them. I do not want to change where I go for my eye doctor. I don't want to do nothing. If I can't go to them, I'll pay for it myself," (Participant 4, Severe DR). While not all participants described longitudinal relationships with specific providers, the narratives suggest that interpersonal relationships and trust may serve as important facilitators of screening adherence within the broad *in-clinic experience* theme.

**Cues to action.** Participants identified various cues that prompted them to seek eye care. These included reminders from office staff and mobile apps. One participant with severe DR shared: "You know, they call you a day or so ahead of time and let you know, you have an appointment," (Participant 2, Severe DR). Similarly, an under-adherent participant said: "I do have the color note app though, so at any point in time I just immediately put it in there," (Participant 2, Under-Adherent). For some participants, personal experiences were a strong motivator to prioritize eye care. One participant with severe DR said: "Because of the experiences I've had with myself, and I'm really concerned about myself. So, I make sure I get to the eye doctor," (Participant 4, Severe DR).

**Knowledge-creating experiences.** Participants reported that learning about diabetes motivated them to seek eye care. Many learned from and shared knowledge with family members. One participant with severe DR said: "But after that, learning how to control the pancreas, the fluid, how to control myself in food…I'm the expert. My sisters, I'm helping them when their diabetes is out of control," (Participant 4, Severe DR). Similarly, one under-adherent participant said: "My granddad had diabetes. And what ended up happening to him, he had to do dialysis, at the end of the day. I'm trying not to do that" (Participant 3, Under-Adherent).

Other participants sought knowledge from diverse sources. One participant with severe DR credited online resources: "Thanks to Internet. YouTube as well…they have a lot of good information about it. College, universities, they have real facts and education," (Participant 3, Severe DR). Another under-adherent participant received information on diabetes from their primary care physician: "I had spoken to my primary care and then I read all the pamphlets that they give you," (Participant 2, Under-Adherent).

## Discussion

Annual eye exams are recommended for early detection of DR as it often presents without symptoms in the early stages [16]. This study validates and expands upon a previously developed framework of DR screening adherence by applying it to two participant populations: individuals with severe DR and individuals who are under-adherent to DR screening [12]. The strong thematic consistency observed across these populations supports the robustness of the framework and

suggests that common individual and structural factors shape engagement behaviors across the disease spectrum. In addition, the *patient-doctor relationship* emerged as a key sub-theme influencing adherence, suggesting that strong interpersonal relationships with healthcare provides may help some individuals overcome otherwise similar structural barriers.

An important theme was how acute changes in vision served as a motivator to seek eye care. Participants often did not perceive a need for screening if their vision remained stable. This is consistent with our previous study, which identified acute vision changes and the need for new glasses as strong motivators for seeking eye care [12]. Similarly, Chou et al. found that "no need" was one of the most frequently cited reasons for not obtaining DR screening within the past year [17]. This reinforces the importance of addressing patient perceptions of need as a critical factor in promoting routine DR screening.

The perceived lack of need for eye care when vision remains adequate highlights a significant gap in our education to patients about DR. Routine DR screenings should be done prior to symptom onset since DR is often asymptomatic in the early stages. The onus of patient education should not be on the patients themselves, but instead is the responsibility of individual providers and the healthcare system. When we asked participants where they learned about diabetic eye disease and screenings from, few cited individual providers or clinic resources such as pamphlets or educational videos and instead cited online resources and learning from the experiences of loved ones. This is consistent with our previous study, where few participants mentioned learning that diabetes could affect vision from their healthcare providers [12]. Similarly, Nwanyanwu et al found that 70.1% of individuals who demonstrated photographic evidence of retinopathy were unaware of their condition [10]. These results align with other studies [18–23], which suggests a substantial need for the healthcare system and individual providers to improve communication about the importance of DR screening to patients before they experience visual changes.

Resource availability emerged as a significant factor influencing screening access. Participants cited transportation issues and lack of insurance as the most significant barriers. Transportation issues were particularly burdensome because participants were reluctant to rely on others for assistance. Similarly, lack of insurance required many participants to pay out of pocket for eye care. Some participants also expressed uncertainty about whether their existing insurance would cover DR screenings. These findings align with previous studies, which highlight cost-related delays and difficulties traveling to appointments as common barriers to eye care [7,17,24–27].

To address resource barriers, telemedicine, which expanded significantly during the COVID-19 pandemic, can partially alleviate transportation issues by allowing patients to access care remotely [28]. Expanding awareness and access to healthcare mobility services, such as ride-sourcing options, is another promising strategy [29,30]. We can also use community-based participatory research to simultaneously identify barriers to care and develop targeted interventions to address these barriers. For example, Frimpong et al previously used community-based focus groups to design a culturally relevant and practical digital health tool for individuals with diabetes based on the needs of people in these focus groups [31].

Interestingly, participants in our prior study did not identify transportation issues as a significant barrier to DR screening despite participants in both studies being from relatively the same geographical area [12]. This discrepancy may be attributed to differences in recruitment settings. We recruited participants in our previous study in a location with co-located ophthalmology and primary care community-based clinics, which likely reduced the impact of transportation barriers. This highlights how the organization of healthcare services can influence the accessibility of care.

Participants consistently described structural barriers such as insurance instability and lack of transportation. However, alongside these barriers, participants also identified facilitators that supported engagement, namely trusting and empathetic patient-doctor relationships. These findings suggest that while structural barriers exist for most individuals accessing DR screening, relational and interpersonal supports may help individuals navigate or persist despite these challenges. This is also supported by other studies, which have shown that high-quality communication and cultural humility in

patient-doctor interactions can significantly increase screening adherence [32]. Similarly, a lack of trust in providers is a barrier to eye care shown in other studies [18].

To improve DR screening access, innovative strategies must be implemented to effectively communicate information, build positive patient-doctor relationships, and address resource barriers. Clinical systems should prioritize educating patients about the link between diabetes and vision loss and emphasize the importance of routine screening. This can be achieved through clinician-reviewed or developed educational videos, which have been shown to improve patient understanding and promote long-term adherence to treatment [33,34]. Other strategies to improve education about DR include patient-centered communication techniques, such as asking open-ended questions, expressing empathy, and sitting face-to-face, which can not only aid in effectively communicating information but are also essential in building positive patient-doctor relationships [35–37]. Strengthening this relationship is key to improving DR screening adherence and can also be achieved by measures such as mitigating time limitations during appointments, mutual agreement and valuing of patient goals, and improving patient perception of clinician expertise [38–41].

When implementing screening protocols, it is also imperative to establish reliable follow-up pathways to ensure timely management and treatment of sight-threatening conditions. Many of the barriers to DR screening identified in this study, such as medical comorbidities, lack of a consistent healthcare provider, and socioeconomic factors, are also known to affect adherence to follow-up care [42–46]. Addressing these interconnected barriers is critical for improving long-term adherence to DR care.

## Study limitations

The strengths of this study include the qualitative approach, which allows for an in-depth and nuanced exploration of participants' experiences. However, several limitations should be noted. Transferability to broader populations may be limited due to a small number of participants being interviewed in a single geographic area. In addition, explicit inclusion of more diverse or atypical participant populations from varied socioeconomic and cultural backgrounds may have yielded additional codes and themes. To enhance transferability, we provide detailed descriptions of the research setting and participant populations, allowing individuals to assess the relevance of the findings to their populations of interest.

This study is also limited as only one researcher conducted all interviews. To address this limitation, we employed post-interview debriefing sessions to guide subsequent interviews and used researcher triangulation during coding analysis with three independent coders including a trained community partner (A.N.). Data triangulation in this study is also limited, though partially mitigated by the inclusion of two participant cohorts. Future studies could further address these limitations by recruiting participants from multiple institutions or geographical areas, involving multiple interviewers, and incorporating perspectives of family members, ophthalmologists, and other healthcare professionals.

Finally, this study is subject to potential researcher bias, as question framing and interactions with participants may potentially influence responses, particularly given that a single researcher conducted all interviews. While we partially mitigated this limitation through researcher triangulation and involvement of a community partner in coding analysis, it is possible that participants' responses were shaped by the researcher's presence or the framing of prompts. Furthermore, early exposure to the pre-existing framework during the interviews may have shaped participants' responses despite the use of open-ended prompts. Similarly, social desirability bias may have led participants to over-report positive attitudes toward care, especially when discussing sensitive topics such as fear of blindness or relationships with healthcare providers. These biases could have influenced the reporting and interpretation of themes, particularly around patient-doctor relationships.

## Conclusions

Further research is needed to better understand DR screening adherence and develop interventions to overcome potential barriers. Further exploration of the patient-doctor relationship sub-theme is warranted to understand patients' comfort

in seeking disease-related information from providers and how this influences adherence behaviors. By examining barriers and facilitators to DR screening across two participant populations, this study strengthens and refines a preciously developed framework of screening adherence. The consistency of themes across differing levels of disease severity and engagement supports the framework's applicability across the care continuum. In particular, the identification of the patient-doctor relationship as a key sub-theme highlights the importance of interpersonal trust and communication in promoting sustained engagement. Future interventions should therefore address both structural barriers and interpersonal facilitators to improve screening adherence among populations at high risk for vision loss.

## Supporting information

**S1 Text. Interview slide deck.**
(PDF)

**S2 Text. Interview transcripts with coding stripes.**
(DOCX)

**S3 Text. Node summaries with coded transcript excerpts by theme.**
(DOCX)

**S1 Table. Qualitative codebook.**
(XLSX)

## Acknowledgments

We would like to thank all the participants for their time and openness in sharing their experiences with us and our community partners for their valuable insights.

## Author contributions

**Conceptualization:** Joana Andoh, Elizabeth Fairless, June Weiss, Kristen Nwanyanwu.

**Data curation:** Joana Andoh, Elizabeth Fairless, Kristen Nwanyanwu.

**Formal analysis:** Julia Fu, Joana Andoh, Althea Norcott, Kristen Nwanyanwu.

**Funding acquisition:** Kristen Nwanyanwu.

**Investigation:** Kristen Nwanyanwu.

**Methodology:** Elizabeth Fairless.

**Project administration:** June Weiss.

**Resources:** June Weiss.

**Supervision:** Kristen Nwanyanwu.

**Writing – original draft:** Julia Fu.

**Writing – review & editing:** Julia Fu, Joana Andoh, Kristen Nwanyanwu.

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
