## [Decision Letter · Decision Letter 0]

17 Dec 2025

PGPH-D-25-02204

Interrogating a framework for diabetic retinopathy screening adherence: qualitative interviews of a severe disease and unengaged population

Dear Dr. Nwanyanwu,

Thank you for submitting your manuscript to PLOS Global Public Health. After careful consideration, we feel that it has merit but does not fully meet PLOS Global Public Health’s publication criteria as it currently stands. Therefore, we invite you to submit a revised version of the manuscript that addresses the points raised during the review process.

The manuscript has been evaluated by three reviewers, and their comments are available below and in the attached document.

The reviewers have raised a number of major concerns. The reviewers raise concerns related to the methodology, and request clarification in the presentation of study findings, further detail regarding the context of the study and further discussion of the study’s limitations.

Could you please carefully revise the manuscript to address all comments raised?

We look forward to receiving your revised manuscript.

Kind regards,

Ilse Bloom

Staff Editor

Journal Requirements:

Additional Editor Comments (if provided):

Reviewers' comments:

Reviewer's Responses to Questions

**Comments to the Author**

1. Does this manuscript meet PLOS Global Public Health’s publication criteria? Is the manuscript technically sound, and do the data support the conclusions? The manuscript must describe methodologically and ethically rigorous research with conclusions that are appropriately drawn based on the data presented.? Is the manuscript technically sound, and do the data support the conclusions? The manuscript must describe methodologically and ethically rigorous research with conclusions that are appropriately drawn based on the data presented.

Reviewer #1: Partly

Reviewer #2: Yes

Reviewer #3: Yes

2. Has the statistical analysis been performed appropriately and rigorously?

Reviewer #1: N/A

Reviewer #2: Yes

Reviewer #3: Yes

3. Have the authors made all data underlying the findings in their manuscript fully available (please refer to the Data Availability Statement at the start of the manuscript PDF file)?

The PLOS Data policy requires authors to make all data underlying the findings described in their manuscript fully available without restriction, with rare exception. The data should be provided as part of the manuscript or its supporting information, or deposited to a public repository. For example, in addition to summary statistics, the data points behind means, medians and variance measures should be available. If there are restrictions on publicly sharing data—e.g. participant privacy or use of data from a third party—those must be specified.requires authors to make all data underlying the findings described in their manuscript fully available without restriction, with rare exception. The data should be provided as part of the manuscript or its supporting information, or deposited to a public repository. For example, in addition to summary statistics, the data points behind means, medians and variance measures should be available. If there are restrictions on publicly sharing data—e.g. participant privacy or use of data from a third party—those must be specified.

Reviewer #1: Yes

Reviewer #2: Yes

Reviewer #3: No

4. Is the manuscript presented in an intelligible fashion and written in standard English?

Reviewer #1: Yes

Reviewer #2: Yes

Reviewer #3: Yes

Reviewer #1: This is an important and interesting paper and is also well written but has several methodological flaws which are described in the attached document. I have provided detailed comments regarding the manuscript which includes comments on the Title, Introduction, Methods, Results and Discussion.

Reviewer #2: I appreciate the authors for conducting this study and enumerating all the factors associated with diabetic retinopathy screening adherence in detail. All the important factors have been explained thoroughly and these are very important for planning any program on Diabetic retinopathy screening. The limitations are a small sample size and participants from a single geographical area.

Reviewer #3: This study by Julia Fu and associates investigates the factors influencing adherence to diabetic retinopathy (DR) screening among individuals with severe disease who underwent interventions and those unengaged in eye care (for over a year), based on qualitative interviews. It was conducted from March 2021 to February 2022. In that period, 8 participants were interviewed once (for approx. 60 minutes) over phone or web based videoconferencing software. The recorded transcripts were analysed to identify the individual, interpersonal, and institutional factors affecting DR screening adherence. Strengths –

1. Qualitative Approach: The study utilized thorough methods, such as grounded theory and the constant comparative technique, giving a detailed perspective on the participants’ experiences.

2. Focus on Vulnerable Populations: The inclusion of individuals with severe DR and those unengaged in eye care provides valuable insights into the barriers faced by high-risk populations.

3. Validation of Previous Framework: Building on an earlier framework, the researchers introduced new interpersonal aspects, e.g., the patient-doctor relationship.

Weaknesses –

1. Small Sample: The study included only eight participants, which may limit the generalizability of the findings.

2. Geographic Scope: It was conducted in a single geographic area, which may not represent broader populations. Also, the racial characteristics conform to a single type.

3. Potential Bias: The qualitative nature of the study introduces the possibility of researcher bias and social desirability bias in participant responses.

Revisions required:

• The authors must comply with the journal’s data-sharing policy by providing access to the analysis of transcripts and coding data to ensure transparency and reproducibility.

• Address the limitations more thoroughly, including the small sample size and geographic scope, and discuss how future research could address these issues.

• Clarify the potential impact of researcher bias and social desirability bias on the findings.

Despite these limitations, the study contributes meaningfully to the understanding of DR screening adherence and offers practical solutions to address barriers. This study is a valuable addition to the literature on diabetic retinopathy screening adherence and is recommended for publication after addressing the minor revisions.

**Do you want your identity to be public for this peer review?** For information about this choice, including consent withdrawal, please see our Privacy Policy..

Reviewer #1: **Yes:** Dr. Shuba KumarDr. Shuba KumarDr. Shuba KumarDr. Shuba Kumar

Reviewer #2: **Yes:** Vignesh T PVignesh T PVignesh T PVignesh T P

Reviewer #3: No

---

## [Decision Letter · Decision Letter 1]

15 Feb 2026

PGPH-D-25-02204R1

Interrogating a framework for diabetic retinopathy screening adherence: qualitative insights from a severely affected and under-adherent population

Dear Dr. Nwanyanwu,

Thank you for submitting your manuscript to PLOS Global Public Health. After careful consideration, we feel that it has merit but does not fully meet PLOS Global Public Health’s publication criteria as it currently stands. Therefore, we invite you to submit a revised version of the manuscript that addresses the points raised during the review process.

We look forward to receiving your revised manuscript.

Kind regards,

Helen Howard

Staff Editor

Journal Requirements:

Additional Editor Comments (if provided):

Reviewers' comments:

Reviewer's Responses to Questions

**Comments to the Author**

Reviewer #1: All comments have been addressed

Reviewer #2: All comments have been addressed

Reviewer #3: All comments have been addressed

publication criteria? Is the manuscript technically sound, and do the data support the conclusions? The manuscript must describe methodologically and ethically rigorous research with conclusions that are appropriately drawn based on the data presented.? Is the manuscript technically sound, and do the data support the conclusions? The manuscript must describe methodologically and ethically rigorous research with conclusions that are appropriately drawn based on the data presented.

Reviewer #1: Partly

Reviewer #2: Yes

Reviewer #3: Yes

3. Has the statistical analysis been performed appropriately and rigorously?

Reviewer #1: N/A

Reviewer #2: Yes

Reviewer #3: Yes

4. Have the authors made all data underlying the findings in their manuscript fully available (please refer to the Data Availability Statement at the start of the manuscript PDF file)?

The PLOS Data policy requires authors to make all data underlying the findings described in their manuscript fully available without restriction, with rare exception. The data should be provided as part of the manuscript or its supporting information, or deposited to a public repository. For example, in addition to summary statistics, the data points behind means, medians and variance measures should be available. If there are restrictions on publicly sharing data—e.g. participant privacy or use of data from a third party—those must be specified.requires authors to make all data underlying the findings described in their manuscript fully available without restriction, with rare exception. The data should be provided as part of the manuscript or its supporting information, or deposited to a public repository. For example, in addition to summary statistics, the data points behind means, medians and variance measures should be available. If there are restrictions on publicly sharing data—e.g. participant privacy or use of data from a third party—those must be specified.

Reviewer #1: Yes

Reviewer #2: Yes

Reviewer #3: Yes

5. Is the manuscript presented in an intelligible fashion and written in standard English?

Reviewer #1: Yes

Reviewer #2: Yes

Reviewer #3: Yes

Reviewer #1: Have uploaded my comments

Reviewer #2: All the reviewer's comments have been addressed.

Reviewer #3: (No Response)

**Do you want your identity to be public for this peer review?** For information about this choice, including consent withdrawal, please see our Privacy Policy..

Reviewer #1: **Yes:** Shuba KumarShuba KumarShuba KumarShuba Kumar

Reviewer #2: **Yes:** Vignesh T PVignesh T PVignesh T PVignesh T P

Reviewer #3: **Yes:** Anand Singh BrarAnand Singh BrarAnand Singh BrarAnand Singh Brar

---

## [Decision Letter · Decision Letter 2]

18 Mar 2026

Interrogating a framework for diabetic retinopathy screening adherence: qualitative insights from a severely affected and under-adherent population

PGPH-D-25-02204R2

Dear Dr. Nwanyanwu,

We are pleased to inform you that your manuscript 'Interrogating a framework for diabetic retinopathy screening adherence: qualitative insights from a severely affected and under-adherent population' has been provisionally accepted for publication in PLOS Global Public Health.

Best regards,

Julia Robinson

Executive Editor

Reviewer Comments (if any, and for reference):

Reviewer's Responses to Questions

**Comments to the Author**

Reviewer #1: All comments have been addressed

Reviewer #2: All comments have been addressed

Reviewer #3: All comments have been addressed

publication criteria? Is the manuscript technically sound, and do the data support the conclusions? The manuscript must describe methodologically and ethically rigorous research with conclusions that are appropriately drawn based on the data presented.? Is the manuscript technically sound, and do the data support the conclusions? The manuscript must describe methodologically and ethically rigorous research with conclusions that are appropriately drawn based on the data presented.

Reviewer #1: Yes

Reviewer #2: Yes

Reviewer #3: Yes

3. Has the statistical analysis been performed appropriately and rigorously?

Reviewer #1: N/A

Reviewer #2: Yes

Reviewer #3: Yes

4. Have the authors made all data underlying the findings in their manuscript fully available (please refer to the Data Availability Statement at the start of the manuscript PDF file)?

The PLOS Data policy requires authors to make all data underlying the findings described in their manuscript fully available without restriction, with rare exception. The data should be provided as part of the manuscript or its supporting information, or deposited to a public repository. For example, in addition to summary statistics, the data points behind means, medians and variance measures should be available. If there are restrictions on publicly sharing data—e.g. participant privacy or use of data from a third party—those must be specified.requires authors to make all data underlying the findings described in their manuscript fully available without restriction, with rare exception. The data should be provided as part of the manuscript or its supporting information, or deposited to a public repository. For example, in addition to summary statistics, the data points behind means, medians and variance measures should be available. If there are restrictions on publicly sharing data—e.g. participant privacy or use of data from a third party—those must be specified.

Reviewer #1: Yes

Reviewer #2: Yes

Reviewer #3: Yes

5. Is the manuscript presented in an intelligible fashion and written in standard English?

Reviewer #1: Yes

Reviewer #2: Yes

Reviewer #3: Yes

Reviewer #1: The authors have adequately and satisfactorily addressed all comments raised earlier. I do not have any further issues

Reviewer #2: All comments have been addressed.

Reviewer #3: All the reviewer's comments were addressed.

**Do you want your identity to be public for this peer review?** For information about this choice, including consent withdrawal, please see our Privacy Policy..

Reviewer #1: **Yes:** Shuba KumarShuba KumarShuba KumarShuba Kumar

Reviewer #2: **Yes:** VIGNESH T.P.VIGNESH T.P.VIGNESH T.P.VIGNESH T.P.

Reviewer #3: **Yes:** Anand Singh BrarAnand Singh BrarAnand Singh BrarAnand Singh Brar
